# Assessing the Outcome of Rehabilitation after Hip Fracture with a Wearable Device—A Study Protocol for a Randomized Control Trial in Community Healthcare

**DOI:** 10.3390/ijerph181910165

**Published:** 2021-09-27

**Authors:** Eva Ekvall Hansson, Agneta Malmgren Fänge, Cecilia Rogmark

**Affiliations:** 1Department of Health Sciences, Lund University, 221 00 Lund, Sweden; agneta.malmgren_fange@med.lu.se; 2Department of Orthopaedics, Skåne University Hospital, Lund University, 205 02 Malmö, Sweden; cecilia.rogmark@skane.se

**Keywords:** hip fracture, postural sway, balance, rehabilitation, wearable device

## Abstract

Background: The increase of the aging population is a challenge to society, as age is related to dependence. Injuries such as hip fractures cause morbidity, loss of independent life, and mortality. The purpose of this protocol is to describe a randomized control trial, with three intervention arms, aiming at investigating if there are any differences in outcomes after hip fracture between different rehabilitation interventions including (1) High-Intensity Functional Exercise (HIFE), (2) HIFE with the addition of continuous measures of movement and body positions with a wearable device, or (3) standard rehabilitation. A secondary aim is to evaluate physiotherapists’ satisfaction with using the wearable device in rehabilitation. Method: Patients with hip fracture that require rehabilitation at home will be invited to participate and randomly assigned to one intervention arm. The primary outcome is balance, measured by postural sway using an Inertial Measurement Unit and by Functional Balance test for Geriatric patients. Secondary outcomes are functional independence in everyday activities, measured with the Barthel Index, and health-related quality of life measured with EuroQol 5 Dimension questionnaire and EuroQol Visual Analogue Scale for health and user satisfaction measured by the User Satisfaction Evaluation Questionnaire. Discussion: This study protocol is the first step in securing the research process before performing a full randomized controlled trial. The next step will be a pilot- and feasibility study.

## 1. Introduction

Even if the incidence of hip fractures is no longer increasing in the Western world, fracture care and rehabilitation will still be a future challenge for the healthcare system [1] due to longer life expectancy, and thereby more individuals will be at risk for hip fracture [2,3]. Many of the risk factors associated with sustaining a hip fracture, such as age, previous fractures, inactive lifestyle, dementia, neuromuscular dysfunction, and malnutrition [4], will also interfere with rehabilitation and thus call for specialized efforts [5]. In addition, the trauma and surgery following a hip fracture can trigger deterioration in terms of delirium, depression, and infections [6,7]. Reduced balance and fear of falling both cause and follow a fracture and need to be addressed in rehabilitation [8]. The risk of new fractures is particularly high immediately after a fracture, which calls for interventions to prevent future fractures [9]. Additionally, the patient’s perception of the rehabilitation process is important for reaching the goals [10]. There is a lack of randomized controlled trials (RCT’s) studying the effects of rehabilitation after a hip fracture [11]. Additionally, the need for using novel methods when evaluating the effect of rehabilitation, such as data captured with wearable devices, has been acknowledged [12]. Examples of such data are the number of steps per day, gait flexibility, and balance [13,14,15]. The use of modern technology might benefit in this [15].

The use of tailored intervention protocols towards patients’ needs and capabilities is recommended, with a focus on balance and proprioceptive endurance together with physical and mental independence [16]. A patient-centered intervention is needed, as individuals with a hip fracture will differ in many aspects, such as pre-fracture medical history and activity, type of surgery, and capability to recover [5,16]. The High-Intensity Functional Exercise (HIFE) program was developed specifically for frail older patients with hip fracture [17], aiming at improving muscle strength, balance, gait, and mobility [18]. HIFE is a person-centered intervention protocol, with exercises chosen for each individual after assessment by a physiotherapist. The program is regularly adapted during the rehabilitation period. HIFE has been evaluated among persons with dementia [19] and among persons living in nursing homes [20], but has to our knowledge not been evaluated when applied to rehabilitation in community health care.

RCT’s studying the effect of rehabilitation after hip fracture are few [11], and their external validity has been questioned as they are performed within a strict protocol [21]. Hence, RCT’s applying tailored intervention protocols, such as HIFE, might increase the usefulness of results from such an RCT in clinical practice.

To evaluate the effectiveness and reliability of rehabilitation programs, a variety of tests and scales has been used in the past, such as Functional Balance for Geriatric Patients (FBG) [22,23] and the Romberg test [24]. Increased postural sway is an important fall risk and can therefore be of interest when assessing effect of rehabilitation after hip fracture [25]. Postural sway can be measured in the individual’s home with wearable devices, a measurement that until now demanded a laboratory environment [13]. However, other outcomes can be of importance for evaluating the effect of rehabilitation programs. Walking speed has a relation to fall risk as well as mortality [26,27], step length has a relation to fall risk [26], and physical activity has a very strong relation to overall health [28]. Wearable devices can monitor physical activity, gait, and position of the body 24 h per day and also provide specific information regarding step length and walking speed [15]. Therefore, they are suitable for individualizing treatment programs [12]. Finally, wearable devices will render feedback to the physiotherapist, in terms of body positions and movements, and help the physiotherapist to adjust intensity and content of the rehabilitation program. We believe this will increase the external validity of the planned RCT’s, better mimicking a real-world setting.

We present a study protocol of an RCT with three intervention arms. The aim of the trial is to investigate if there are any differences in outcomes of rehabilitation after hip fracture between different rehabilitation interventions according to (1) HIFE, (2) HIFE with the addition of continuous measures of movement and body positions with a wearable device, or (3) standard rehabilitation in community healthcare. A secondary aim is to evaluate how satisfied physiotherapists are with using the wearable device in the rehabilitation process and if there are any differences between the three interventions in respect of the amount of home visits or telephone calls by the physiotherapist.

Our hypothesis is that HIFE and continuous measures of movement and body positions with a wearable device will have positive effects on balance, functional independence in everyday activities, and health-related quality of life compared with standard rehabilitation or HIFE without continuous measures. We also hypothesize that continuously measuring movements and body positions will support the physiotherapist in planning further contact with the person and that user satisfaction (the physiotherapist) will be high.

## 2. Materials and Methods

This is a study protocol of an RCT with three arms: two interventions and one control.

### 2.1. Setting

The study will be performed in Malmö, which is the third-largest city in Sweden, with about 330,000 inhabitants. In Malmö, about 700 persons are treated for hip fractures every year. We estimate that about 25% of these will fulfill the inclusion criteria for this study and that it therefore will take about 12 months to allocate the 144 participants needed.

### 2.2. Recruitment and Inclusion/Exclusion

Inclusion criteria will be patients at any age who are discharged from hospital after treatment for a hip fracture, living in Malmö, and who are unable to visit a rehabilitation facility and therefore request treatment at home. Another inclusion criterion is the ability to read and understand the Swedish language or the ability to understand information through an interpreter. Since major neurological diseases that have an impact on balance and walking ability will affect the possibility to perform exercises in the HIFE program, the presence of such a disease is an exclusion criterion. Since we expect that persons with a diagnosed cognitive disease, or moderate to severe cognitive impairment will need more assistance from caretakers to perform the HIFE program and to use the wearable device than the project has possibility to provide, this is also an exclusion criterion and will be assessed by a physiotherapist (PT).

Persons fulfilling the inclusion-criteria will be informed about the study at the first home visit by the PT. Those who accept to participate provide their written consent and a second home visit is scheduled.

### 2.3. Instrumentation

An Inertial Measurement Unit (IMU) measures linear accelerations and rotational velocities, often combined with a GPS to reduce the drift in the position level [28,29]. Two different IMU’s will be used in the study: one for measuring postural sway and one for measuring body positions and movements. An IMU containing algorithms for measuring postural sway will be used in all three intervention arms and obtain measures of postural sway in the anterior-posterior and medio-lateral direction at baseline and at follow ups. The IMU used has shown good validity and reliability for measuring postural sway [13]. When measuring postural sway, the IMU is attached to the lower back and postural sway is measured when the participant is standing still, for 30 s. Another IMU, containing algorithms for measuring body positions and movements will be used in one of the intervention arms (intervention B). Body positions are time spent sitting, time spent standing or walking, time spent in supine, prone, and side lying. Movements measured are strides, step time, falls, near falls, and variation in gait. The algorithms in this IMU were developed within the Modern Technology against Falls (MoTFall) project and have shown good sensitivity and specificity for measuring a near fall [15]. When used in the intervention, the participant wears the IMU on the right thigh, approximately 10 cm above the knee, under the clothes, attached with skin-friendly adhesives. The IMU is worn 24 h per day and is charged twice a week. Data from the IMU is sent through a router mounted in the participant’s home, to a platform, where the PT can obtain the data simultaneously and follow the participant’s progress in the rehabilitation process. Data collected during time when the participant is too far away from the router is saved in the IMU and uploaded to the platform when the IMU is charged.

### 2.4. Interventions

In order to increase the usefulness of the results from this RCT, all three groups receive interventions that are tailored to each participant, after assessment by a PT, either within the specified frame of standard rehabilitation (control group) or HIFE (intervention A and B). Additionally, the rehabilitation time periods are tailored to each participant in order to adjust the study to a clinical setting. However, a minimum number of sessions as well as a time interval for the rehabilitation period will be defined after analysis of the results collected in the planned pilot and feasibility study.

Control group: the intervention comprises the standard rehabilitation services: walking training and adaptions to the walking pattern of the patient prior to surgery together with functional exercises [30]. The intervention is tailored to each patient, based on the type of fracture, type of surgery, and assessment by the PT as well as treatment goals.

Intervention A: the intervention comprises the HIFE protocol and, if necessary, other additional exercises. HIFE consists of five exercise categories depending on the capabilities of the patient. A task of walking a short distance (5–10) meters without aid works as a guideline for the PT to select the most suitable exercises for each individual. The five exercise categories and exercises included in each category are shown in Table 1. The PT selects the timetable of the session and creates a progressive high-intensity program based on the pace of the patient. There are three intensity scales for both strength exercises, which depend on the sets of Repetition Maximum (RM) and the balance exercises where postural stability may be challenged close to the limit of maintaining postural stability (Table 2). A 5-min warmup of lower and upper limbs is applied while sitting before the balance exercises begin. Patients start the warmup with walking on the spot, opposing arm swings at the side of the body, “sewing-machine” steps, “picking apples” movement towards different directions, knee stretches, and steps to the side and back of both legs alternately. The equipment needed for the sessions is portable and consists of step boards and chair cushions (5 cm height min), weighted belts (1 kg), soft pads, mattresses and balls, bean bags, belts with handles, and chairs without arm support. All categories have various exercises for the PT to select (Table 1). Difficulty can be increased by providing less assistance or use softer surfaces. Strength intensity is enhanced with weight belt on A and D categories [18]. HIFE has been shown to improve balance, gait, and muscle strength among older persons who were dependent in activities of daily life [31].

Intervention B: the intervention comprises the HIFE rehabilitation protocol, and, if found necessary by the treating PT, other additional exercises. In addition, follow-up and feedback during the rehabilitation period will be given via an IMU, which participants wear 24 h per day until the rehabilitation period has ended (i.e., treatment goals are achieved). The treating PT is thereby provided with specific information regarding time spent per day in sitting, lying, and standing, the total amount of steps per day, any changes in walking speed, etc. Hence, follow-up and feedback can therefore include specific information from the PT to the patient regarding the progress of rehabilitation.

### 2.5. Outcomes

The primary outcome is balance in terms of postural sway and functional balance. Measures of postural sway are performed with the patient standing with heels approximately 10 cm apart and feet pointing about 30° outwards for 30 s with eyes open (EO) and then 30 s with eyes closed (EC). During the measure, an IMU is mounted on the patient’s lower back, in level with L5. The IMU has shown good validity and reliability for measuring postural sway compared to the golden standard (force plate) [13]. Functional balance is assessed with the Functional Balance test for Geriatric patients (FBG), and grades the performance with points, where 0 stands for no execution and 6 for the higher level of performance. FBG evaluates the ability of the patient to sit and stand up, maintain a standing position, walk, and turn. The test includes four activities that the patient shall complete, and every activity has 7 levels of difficulty, indicating the level of independence in each task. The maximum points achievable is 24 [32]. The FBG has shown good validity and reliability when compared to the golden standard (Berg Balance Scale) [22].

Secondary outcome measures are functional independence in everyday activities, health-related quality of life, and the physiotherapist’s satisfaction with the IMU.

Functional independence in everyday activities will be assessed using The Barthel Activity of Daily Living questionnaire. Participants are asked how they manage various everyday activities such as food intake, locomotion, toilet visits, dressing, bathing, transferring, and continence. Ten activities are included, rated on a three-graded scale 0–2 with 0 indicating the worst condition/state and 2 the best condition/state. The maximum score is 20 [33].

Health-related quality of life will be measured by the EuroQol 5-Dimension Questionnaire (EQ5D). EQ5D is a generic instrument to measure mobility, self-care, everyday activities, pain, and anxiety, with five response alternatives, ranging from no problem to unable/extreme [34]. The EQ5D also consists of a visual analog scale, where the participant is asked to rate their health on a vertical scale, where 100 is the best imaginable health state, and 0 is the worst imaginable health state. EQ5D is widely used for monitoring health status among different patient groups all over the world [35].

User satisfaction will be assessed with the questionnaire User Satisfaction Evaluation Questionnaire (USEQ). USEQ consists of six questions where users fill in how satisfied they are with using a system in rehabilitation on a score from 1 (not at all) to 5 (very much). Since it is PTs who fill in the questionnaire by the end of intervention B in this study, the sentence “in your work” was added to the questions as well as “information from the system” instead of only “the system”. The total score lies from 6 (min) to 30 (max). To compute the score, all questions are considered positive except Q5, which is a negative question. Thus, if the person selects 3 in Q1, we add 3 to the total score; on the other hand, for Q5 we subtract the numerical value of the response from 6, and if a person selects 2 in Q5 4 is added to the total value [36].

Information about the number of home visits and telephone calls necessary for the patients to acquire independence is generated through medical records.

### 2.6. Procedure

Background data on age, sex, method of surgery for the hip fracture, and medication will be collected at baseline. All other measures will be collected at baseline and end of the rehabilitation period. Postural sway and balance will also be assessed 3 months after the end of the rehabilitation period. At the end of the rehabilitation period, physiotherapists fill in USEQ and also record how many home visits were performed and time spent at each home visit as well as the number of phone-calls each participant required.

Baseline data, including information about age, sex, method of surgery for the hip fracture, and medication, as well as outcome measures, will be obtained by a PT at a regular home visit. Follow-up measures will be performed at the end of the rehabilitation period, including primary and secondary outcome measures. The rehabilitation period is defined by the needs of the individual participant and can hence differ in time between participants.

At the end of the intervention treatment, the PTs will fill in the USEQ questionnaire regarding how satisfied they were with using the information from the IMU. During the study, PTs will document the duration and number of home visits as well as telephone calls. PTs time and number of visits is not a pre-designed factor, and it will adapt to the patient’s necessities. The treating PT will also take notes regarding participants who discontinue or deviate from the intervention’s protocols, including reasons for discontinuing or deviating.

All PTs in community healthcare in Malmö, who rehabilitate persons with hip fracture in the Malmö municipality, have received education in the HIFE program. Additionally, workshops including an introduction of the IMU, how to use it, and how to retrieve data from it have been performed. Inclusion of participants to the pilot and feasibility study started in May 2021 and is expected to be completed in December 2021. After that, any necessary changes to the study protocol will be made and a decision whether the full RCT should be conducted will be made.

### 2.7. Sample Size

Since balance, measured by postural sway, is the main outcome measure, postural sway was used to determine the statistical power of the study. We used results from a previous study [25] with a mean value of 1.8 mm/s for difference in postural sway in the anterior-posterior (AP) direction between two groups of community-dwelling older persons. Since we want to calculate differences between all three groups, a significance level of 0.017 was determined, giving a necessary sample size of 43 participants in each group to reach a power of 80% [37]. To compensate for dropouts, the sample size was set to 48 participants in each group.

### 2.8. Randomization

To enhance the power of the study, to reduce bias of covariates and ensure equal dispersion of the participants at all times [38], block randomization will be applied through a website [39]. The number of subjects in each block will be 48 and the number of blocks will be 3, generating a randomization plan with subjects randomized in 3 different treatment blocks.

An independent person will perform the randomization process. All baseline measures will be performed before randomization to any of the interventions. Blinding of the assessor is not possible at present, due to the restrictions caused by the Covid-19-pandemic, prohibiting unnecessary home visits. However, when these are lifted, follow-up measures at 3 months will be performed by an independent PT, not aware of which intervention the participants have been allocated to. Blinding of participants is not considered possible.

### 2.9. Statistics

Differences between the three groups will be analyzed. Normally distributed data will be analyzed using ANOVA and on abnormally distributed data, the Kruskal-Wallis test will be applied [37]. The ANOVA test will show if there is a significant difference between the groups and a post-hoc analysis will detect where the difference lies. In all analysis, intention to treat will be used.

### 2.10. Feasibility Study

In order to test the protocol and measure the feasibility of the planned RCT, a feasibility study will be performed before starting the full RCT, using a sample size of 20% of the calculated sample size for the full trial. In the pilot and feasibility study, adverse events, recruitment rate, compliance to the intervention, retention rate, and the ability to collect outcome measures will be tested. If necessary, the protocol will be adjusted according to findings in the feasibility study.

### 2.11. Ethical Considerations

Participants can drop out from the study at any time, without giving any explanation and without affecting their rehabilitation or care.

A de-identification process will be used to preserve the integrity of the participants. The de-identification procedure will be conducted via a code list with names and birth dates of each participant connected to a number. The code list will be stored in a safety box that only the main researcher will have access to.

All information will be stored digitally at a secure server at Lund University, which only the researchers have access to, in accordance with European Union Data Protection Regulation 2016/679 (GDPR) [40] and the Data Protection Act, 2018:218 [41]. The study has been approved by the Swedish Ethical Review Authority (drn 2020-00789).

## 3. Discussion

The interventions included in this study are complex in character [42]. By publishing the study protocol, we aim to secure the research process, and test both the protocol and the feasibility of the study, according to the Medical Research Council’s guidance [43].

There is a need for clinical trials designed to enhance that the knowledge gained from the trial have an impact in clinical practice [44]. Hence, to better apply to clinical practice, the interventions in this planned RCT will be tailored to each participant within the frame of HIFE or within the frame of standard rehabilitation. There is, however, always a risk that components included in an experimental intervention also are included in standard practice [43], which needs to be considered in the data analysis and in the interpretation of the results. HIFE is an intervention method that helps engagement in physical activities, suitable for a population with physical and cognitive impairments that shows no negative impact and provides independence in ADL conditions [17]. Studies comprising HIFE interventions have so far taken place in a controlled environment of residential care facilities [19,45]. Since older persons in residential care are at higher risks of falling compared with community-living older people, a home-based treatment could lead to another outcome [46]. Home-based rehabilitation is often the first choice for many patients due to the inability to travel to a healthcare facility and has also shown to have positive effects on physical functioning [47]. The effectiveness of HIFE in fall prevention, balance, and muscle enhancement have been evaluated, using assessment tools like the Barthel index and Berg Balance Scale [17,19,45]. Evaluation of HIFE using assessment tools targeting balance among older persons, like the FBG test, has, to our knowledge, not been performed earlier, and neither have evaluations using such a specific balance assessment as postural sway been utilized before.

Our study will add new information on the effect of HIFE on balance among older persons, using the FBG test and postural sway. This will add to the formerly known effectiveness of HIFE in fall prevention, balance, and muscle enhancement [17,19,45].

This study protocol does not include any specific measure of muscle strength. However, functional balance, here measured in the FBG test, demands muscle strength and so does functional independence in everyday activities, here measured with the Barthel questionnaire. Strength has a relation to dependence in everyday activities [48]. Baseline measurers will be performed shortly after discharge from the hospital and a measure of muscle strength can therefore be highly influenced by pain. Additionally, we want to limit the total number of tests, since performing them can be physically challenging.

In intervention B, in addition to HIFE, data of body positions and movements are collected using an IMU. Data is collected via the IMU 24 h per day and transferred through an application to the treating PT giving him/her an opportunity to follow the participants’ progression in the rehabilitation process on a day-to-day basis. This also enables the PT to provide constructive feedback to the participants during the monitoring period. Wearable devices with a built-in accelerometer have shown the ability to provide accurate data with individual characteristics [49]. Laboratory, as well as home-based studies, have shown that accelerometer technology can accurately evaluate balance and risks of falls in an older population [50]. However, whether this has any effect on the outcomes of rehabilitation has, to our knowledge, not been studied before. In addition, concerns regarding patient privacy, management of system operation, and the magnitude of data need to be evaluated carefully before the use of these type of devices [51].

The study will evaluate the balance of the participants via postural sway and FBG tests. The FBG test, specifically designed for older people, will evaluate the static and dynamic balance and participant’s gait. Wearable devices make it possible to measure postural sway outside the laboratory setting, and an IMU is easy to transport and easy to use. Thus, the IMU is a novel tool for assessing postural sway in a clinical setting.

Randomization will be performed with block randomization to ensure an equal number of participants in each group. We expect that there might be more dropouts from the intervention including monitoring with an IMU. Therefore, we made a generous calculation of the dropout rate when calculating the sample size.

The next step before performing a full RCT will be to perform a pilot and feasibility study [43] using the study protocol described here. The RCT has been registered in Clin.Trial.gov (2020-00789).

## 4. Conclusions

This study protocol described an RCT, testing a structured procedure for rehabilitation after hip fracture in community care, where two interventions are compared (High-Intensity Functional Exercise and High-Intensity Functional Exercise and monitoring with a wearable device) with control (standard rehabilitation).

## Figures and Tables

**Table 1 ijerph-18-10165-t001:** Categories and examples of HIFE exercises.

Categories	Example of Exercises
A: Static and dynamic balance exercises in combination with lower-limb strength exercises	Squats in different positions, bodyweight transfer, standing up from sitting, lunges.
B: Dynamic balance exercises in walking	Walking with different degrees of difficulty, step over.
C: Static and dynamic balance exercises in standing	Maintaining stance with different degrees of difficulty, turning head in various directions, squats with different starting positions, bodyweight transfer in different positions, reaching for an object, trunk rotation, throwing and catching a ball, sidestep and turn.
D: Lower-limb strength exercises with continuous balance support	Squats in different positions, standing up from sitting in parallel stance, heel raises, bodyweight transfer, walking in stairs.
E: Walking with continuous balance support	Walking on a flat surface, walking in different directions, walking and turning.

**Table 2 ijerph-18-10165-t002:** Intensity scales of HIFE exercises.

Exercise Session	High Intensity	Medium Intensity	Low Intensity
**Strength exercises**	Sets of 8–12 RM	Sets of 13–15 RM	Sets of >15 RM
**Balance exercises**	Postural stability fully challenged *	Postural stability not fully challenged or fully challenged in a minority of the exercises	Postural stability not challenged

* Defined as performed close to the limit of maintaining postural stability.

## Data Availability

Data is available on request to the corresponding author.

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
