# Peer review of "Assessing the Outcome of Rehabilitation after Hip Fracture with a Wearable Device—A Study Protocol for a Randomized Control Trial in Community Healthcare"

_ijerph, 2021, doi:10.3390/ijerph181910165_

Round 1
Reviewer 1 Report
The work by Hansson and colleagues describes a study protocol for a randomized controlled trial aiming to compare three rehabilitation interventions (HIFE + IMU feedback vs. HIFE vs. standard rehabilitation) in individuals after hip fracture.
Reviewer comment 1: The discussion needs to be revised and undergo a thorough spelling check. Several text parts are duplicated, e.g., lines 314-315 vs. 294-295; 326-327 vs. 301-302 (also a frequent error in the manuscript: it should state “studies comprising HIFE interventions have […]”). Please check for further duplicates.
Reviewer comment 2: Since the IMU measurement in addition to the HIFE seems to be the central part of the planned study, I encourage the authors to describe in detail how the data provided by the IMU measurement will be used to adapt the exercise intervention.
Reviewer comment 3: Do the authors have any plans on documenting whether the target intensity during the exercise intervention was achieved? Please describe how this will be done for both strength and balance exercises. Furthermore, even though this might be difficult, a definition of how “fully challenged” is defined would enhance the reproducibility of the planned study.
Reviewer comment 4: I endorse that the authors aim to move away from a “strict controlled trial” towards a “real world setting”, potentially making the findings more applicable for practice. Nonetheless, I will raise two points:
- What is the anticipated number of therapy sessions during the intervention period? How many therapy sessions will approximately be performed per week? The number of sessions performed per individual during the intervention period is a variable that needs to be adjusted for in the statistical analyses, otherwise, it may induce substantial bias. In addition, I recommend that the authors better standardize this parameter across all groups (i.e., setting a minimum number of exercise sessions for each participant within the intervention period and defining the length of the intervention period). This will reduce bias for between-group comparisons and again enhance the reproducibility of the planned study.
- Even though this alone seems not to be sufficient to ignore my prior remark but have the authors thought of performing an intention to treat analysis (i.e., including every randomized participant independent of their compliance with the exercise intervention)? This would support the “real world setting” of the planned study.
Reviewer comment 5: Intervention A consists of HIFE plus feedback about the progress obtained via IMUs. Please explain how exactly this information will be used. Will the PT always consider this information for the guiding of the exercise sessions and will also the participants be regularly informed about their progress?
Reviewer comment 6: I do not think that it makes sense from a physiological standpoint to combine balance and strength training as this will lead to suboptimal improvements in both parameters. Would it not be more favorable to first perform the balance exercises and afterwards the strength exercises? This would lead to a more effective stimulus for both parameters. What is the rationale for choosing a combined intervention?
Reviewer comment 7: Considering that HIFE is a combined strength and balance exercise regime and not only improved balance but also parameters of muscular function (especially rate of force development) may reduce the risk of falls, it could be beneficial to include a measure of muscular strength/capacity as a secondary outcome. This may be as simple as a sit-to-stand test that can easily be implemented at home.
Minor Comments:
- Many sentences are grammatically incorrect. Please perform a thorough spelling check.
- Line 29: I recommend changing the order of the first paragraph. A general sentence about hip fractures would make it easier to jump into the topic. The statement about the lack of RCT’s and studies using wearable technology should be mentioned later on in the paragraph.
- Author contributions: Please remove “For research articles with several authors, a short paragraph specifying their individual contributions must be provided. The following statements should be used”
- Informed consent statement: Change to “Informed consent will be obtained from all subjects involved in the study” or “not applicable”.
Author Response
Thank you for your time and effort to improve our manuscript. Please find authors response in the attached file.

Reviewer 2 Report
Thank you for the opportunity to review the article entitled: Evaluation of the outcome of rehabilitation after hip fracture with a portable device: a study protocol for a randomized control trial in community health care. This work describes a clinical trial protocol to evaluate the rehabilitation effect of three interventions in patients with hip fractures.
I have some questions and comments about this interesting work.
On line 258, it is necessary to add a reference for the web that allowed block randomization.
In point 2.8, it is necessary to reinforce why this randomization will be by blocks.
As described by the authors, after three months, the therapy "will be performed by an independent PT". Will the PTs be the same on all assessments? Is there any bias if the PT is changed? Could the authors control these?
Point 2.9 should be rewritten. The nonparametric tests described to assess the differences between the three groups is not correct. Did the authors not evaluate qualitative variables? It would also be essential to carry out the evaluations and comparisons within the groups.
Are there confusing variables that need to be controlled? If so, specify and mention with which statistical tests will be evaluated.
I believe that the authors should describe the possible problems that may arise in patients and how they will control them. For example, if a patient requires some extra intervention or surgery, will they leave the study? It is necessary to go deeper into these points.
Author Response
Thank you for your time and effort to improve our manuscript. Please find our changes specified in the attached file.

Reviewer 3 Report
Thank you for the opportunity to review your protocol. I believe that this future randomized controlled trial will only increase the scientific knowledge about hip fracture rehabilitation.
My only appreciation is related to the intervention program.
I am aware that it must be individualized to each patient and each clinical situation, but it is also true that monitoring the intensity exclusively with the number of repetitions is too "permissive".
I think that before developing the work, it should be clarified how the intensity is going to be monitored. A) based on % direct 1RM (not recommended); B) based on % indirect 1RM (you've written based on RM repetitions but, I think it is not clear, because in addition each % of 1RM must be assigned to a number of repetitions.; c) based on speed of execution; d) based on the character of effort (number of repetitions of reserve); e) time under tension; f) others.
Author Response

(The authors gave the same response as above.)

Round 2
Reviewer 1 Report
My points raised have been adequately addressed. Yet, I want to use the opportunity to reply to the authors‘ response to reviewer comment 6:
There is no doubt that there will always be a shared contribution of aspects of strength and balance to certain exercises/movements. Yet, it is important to differentiate what the focus of the exercise training is. It most certainly is possible to split an exercise session into one part with a focus on balance and a second part with a focus on muscular strength. Here basic principles of exercise training come into play: 1) principle of specificity as discussed in e.g. https://www.mdpi.com/2076-3417/11/8/3401. It describes that “a training program must stress the physiological systems used to perform a particular activity in order to achieve specific training adaptations. As the aforementioned study showed, strength training on unstable surfaces did not translate to strength gains on stable surfaces. The underlying problem is that unstable surfaces or balance-focused strength exercises will reduce the total load during the training session which is considered one of the most important stimuli for muscle growth and muscular strength. 2) The principle of exercise order aims to reduce the detrimental effect of neuromuscular fatigue. Exercises with the highest priority and that demand the highest level of concentration (balance in the case of the planned study) should be performed at the beginning of the exercise session. That said, I am not arguing against combining strength and balance exercises within the same session but against combining these two aspects in the same exercise. An exercise should either be performed with a full focus on improving muscular strength or balance. I recommend considering this when interpreting the results of the planned study or even for future research of your group on the topic. Best luck with your project.
Author Response
Thank you for those valuable comments. Please find the response in the attached file.
Best regards
Eva Ekvall Hansson
